# Sepsis-associated neuroinflammation in the spinal cord

**Akiko Hirotsu**[ID]**, Mariko Miyao, Kenichiro Tatsumi, Tomoharu Tanaka**[ID] *

Department of Anesthesia, Kyoto University Hospital, Kyoto, Japan

* 665tana@kuhp.kyoto-u.ac.jp

## Abstract

Septic patients commonly present with central nervous system (CNS) disorders including impaired consciousness and delirium. Today, the main mechanism regulating sepsis-induced cerebral disorders is believed to be neuroinflammation. However, it is unknown how another component of the CNS, the spinal cord, is influenced during sepsis. In the present study, we intraperitoneally injected mice with lipopolysaccharide (LPS) to investigate molecular and immunohistochemical changes in the spinal cord of a sepsis model. After LPS administration in the spinal cord, pro-inflammatory cytokines including interleukin (IL)-1β, IL-6, and tumor necrosis factor alpha mRNA were rapidly and drastically induced. Twenty-four-hour after the LPS injection, severe neuronal ischemic damage spread into gray matter, especially around the anterior horns, and the anterior column had global edematous changes. Immunostaining analyses showed that spinal microglia were significantly activated and increased, but astrocytes did not show significant change. The current results indicate that sepsis induces acute neuroinflammation, including microglial activation and pro-inflammatory cytokine upregulation in the spinal cord, causing drastic neuronal ischemia and white matter edema in the spinal cord.

## 1. Introduction

During sepsis, the host's excess immune response can affect various organs [1]. Because of blood–brain barrier (BBB), the central nervous system (CNS) was previously thought to be isolated from systemic inflammation during sepsis [2]. However, recent studies have revealed that patients with severe sepsis commonly report brain-related symptoms such as delirium, seizures, and impaired consciousness even without direct infection in the brain [3, 4]. Those brain abnormalities have been collectively referred to as sepsis-associated encephalopathy (SAE) [5, 6]. Although the exact mechanism of SAE is still controversial, neuroinflammation is now regarded as one of the main factors [7]. In sepsis, brain residual glial cells, especially microglia, are activated in response to systemic inflammation. These cells produce pro-inflammatory cytokines such as interleukin (IL)-1β, IL-6, and tumor necrosis factor alpha (TNFα) [8, 9]. Additionally, the BBB can be damaged by systemic inflammation, with peripheral inflammatory cells infiltrating the brain [10]. Migrating peripheral cells and activated residual glial cells synergistically induce inflammation processes in the brain. This can damage neuronal functions and lead to SAE [11, 12].

**Data Availability Statement:** All relevant data are within the paper and its Supporting Information files.

**Funding:** TT received the Grant-in Aid for Scientific Research (17K11076) from the Japan Society for the Promotion of Science (Tokyo, Japan). The

funder had no role in study design, data collection and analysis, decision to publish, or preparation of the manuscript.

**Competing interests:** The authors have declared that no competing interests exist.

Another component of the CNS, the spinal cord, has rarely been investigated in relation to sepsis. Neuroinflammation in the spinal cord is associated with injury [13], infection [14], and various degenerative diseases [15]. However, research is lacking as to whether neuroinflammation itself occurs in the spinal cord during sepsis, and how those changes could influence spinal functions. It is clinically common to observe impaired motor and sensory functions associated with sepsis [16], but those changes have been considered peripheral nerve and muscle disorders [17, 18]. The spinal cord has a similar anatomical structure to the brain, that is, isolated by the blood-spinal cord barrier (BSCB) and composed of neurons and glial cells [19]. In the present study, we used sepsis model mice to investigate how sepsis could affect the spinal cord molecularly and morphologically.

## 2. Materials and methods

### 2.1. Animals

This study (Permit Number: Med Kyo 19543) was approved by the Animal Research Committee of Kyoto University (Kyoto. Japan). All experiments were conducted according to the institutional and National Institutes of Health guidelines for the care and the use of animals. BALB/c and C57BL6 male mice were purchased from Japan SLC Inc. (Shizuoka, Japan). All experiments were performed with BALB/c, but, for supplementary figure experiments, C57BL6 mice were used. Mice were maintained under a 24˚C, 12-h light/dark controlled environment with *ad libitum* feeding. All surgery was performed under sevoflurane anesthesia, and all efforts were made to minimize suffering. At the time point of specimen collection, mice were euthanized by sevoflurane inhalation followed by cervical dislocation or decapitation.

### 2.2. Drugs and chemicals

Lipopolysaccharides (LPS) from Escherichia coli O55:B5 (L28880) were purchased from Sigma-Aldrich (St. Louis, MO, USA). Sevoflurane (PubChem CID: 5206) was obtained from Mylan Pharmaceutical Co., Ltd. (Osaka, Japan).

### 2.3. Establishment of mice sepsis model

The mouse sepsis model was established by LPS administration or cecal ligation and puncture (CLP). In the LPS-induced sepsis model, LPS or the same amount of saline were intraperitoneally administrated to C57BL6 male mice. The amount of LPS (1.25 or 2.5 mg/kg) was previously determined [20]. CLP was performed as follows: Mice were put in an anesthesia bottle and induced with sevoflurane. The mice were placed in a supine position, maintaining anesthesia by 2% sevoflurane with a nosecone cover on the face. Then, we shaved the abdomen, disinfected the skin with 70% alcohol, and made a midline abdominal incision of approximately 1.5 cm with a surgical knife. The cecum was carefully identified, ligated at 1 cm of its distal end, and sutured with No. 3–0 silk (Alfresa, Osaka, Japan). The tip of the ligated side was punctured using an 18-gauge needle and then replaced into the abdominal cavity. The peritoneum and skin scar were sutured with No. 3–0 and No. 4–0 silk, respectively. The nosecone cover was removed, and the mice were placed in their cage. The sham operation was a skin and peritoneal incision and cecum identification. During the procedure, the rectal temperature was maintained above 37˚C.

### 2.4. Reverse transcription and real-time quantitative polymerase chain reaction (qRT-PCR)

Total RNA was isolated from the spinal cords using a Nucleospin RNA II kit® (Macherey-Nagel, Düren, Germany). First-strand cDNA synthesis and real-time reverse transcription-

polymerase chain reaction (RT-PCR) were conducted with One Step CYBR™ RT-PCR Kit II (Takara Bio, Shiga, Japan) according to the manufacturer's instructions. RT-PCR assays were conducted using the 7300 Real-Time PCR System (Applied Biosystems, CA, USA). The PCR primers of 18S and IL-6 for mice were obtained from Qiagen (Valencia, CA, USA) (Catalog Numbers: 18S mouse; QT02448075, IL-6 mouse; QT00098875). IL-1β, TNFα, COX-2, and IL-36γ were obtained from Invitrogen (CA, USA). Primer sequences were as follows: IL-1β 5′–ATGAGGACATGAGCACCTTC–3′ (forward) and 5′–CATTGAGTTGGAGAGCTTTC–3′ (reverse), TNFα 5′–TCGTAGCAAACCACCAAGTG–3′ (forward) and 5′–CCTTGAAGAGA ACCTGGGAGT–3′ (reverse), COX-2 5′ –TGAGCAAC– TATTCCAAACCAGC–3′ (forward) and 5′ –GCACGTAGTCTTCGATCAC– TATC–3′ (reverse), and IL-36γ 5′ –AGAGTA ACCCCAGTCAGCGTG–3′ (forward) and 5′ –AGGGTGGTGGTACAAATCCAA–3′ (reverse). For each target mRNA, the fold changes in expression were calculated relative to 18S rRNA.

## 2.5. Enzyme-linked immunosorbent assays (ELISA)

Immediately after the euthanasia by decapitation, blood was sampled from the left ventricle. Sampled blood was centrifuged at 3,000 ×g for 15 min, and the supernatants were extracted and frozen at −20˚C until the experiment was conducted. Serum IL-6 concentrations were assayed using an IL-6 ELISA kit (Abcam plc, Cambridge, UK) according to the manufacturer's instructions. Briefly, each 20 μl of serum was diluted fivefold with sample dilution buffer and was applied to IL-6 96-well microplate with concentration-regulated standard solutions. The plate was covered and incubated overnight at 4˚C with gentle shaking. After washing four times with wash solution (300 μl each), 100 μl aliquots of biotinylated IL-6 detection antibody was added and incubated for 1 h at room temperature with gentle shaking. After washing four times, 100 μl aliquots of horseradish peroxidase (HRP)–streptavidin solution were added and incubated for 1 h at room temperature prior to washing four times again. Finally, 100 μl aliquots of 3,3′,5,5′-tetramethylbenzidine (TMB) one step substrate reagent was added to develop blue color. After incubating for 30 min at room temperature in the dark with gentle shaking, 50 μl aliquots of stop solution was added to each well to change the color from blue to yellow. Immediately, absorbance intensity was measured at 450 nm with a reference wavelength of 655 nm.

## 2.6. Histochemical analysis

Immediately after euthanasia by cervical dislocation, mice were transcardially perfused with PBS, followed by 4% paraformaldehyde (PFA). Then, the spinal cords from the upper thorax to the lower lumbar cord were harvested containing surrounding vertebra. The spinal cords were fixed with 4% PFA for 2 days at 4˚C and penetrated into 10% EDTA-Na (pH 7.4) for vertebral decalcification. After decalcification, they were embedded in paraffin as transverse sections (10 μm thick). Then, we stained them with hematoxylin and eosin, followed by immunohistochemistry with anti-neuronal nuclei (NeuN) (MAB377, Merch Millipore, Darmstadt, Germany, diluted to 1:100 in PBS), a specific marker for neurons, anti-glial fibrillary acidic protein (GFAP) (#3670, Cell Signaling, Danvers, MA, USA, diluted to 1:50 in PBS), a specific marker for astrocytes, and anti-ionized calcium binding adapter protein (IBA)-1 antibodies (ab107159, Abcam plc, Cambridge, UK, diluted to 1:2,000 in PBS) a specific marker for macrophage/microglia. Immunohistochemistry analysis was conducted as follows: Endogenous peroxidase activity was blocked by 0.3% $H_2O_2$ in methyl alcohol for 30 min. The glass slides were washed in PBS (six times, 5 min each) and mounted with 1% normal serum in PBS for 30 min. Subsequently, the primary antibodies were applied overnight at 4˚C. They were incubated with biotinylated secondary antibodies diluted to 1:300 in PBS for 40 min, followed

by washes in PBS (six times, 5 min). We then applied the avidin-biotin-peroxidase complex (ABC) (ABC-Elite, Vector Laboratories, Burlingame, CA) at a dilution of 1:100 in bovine serum albumin (BSA) for 50 min. After washing in PBS (six times, 5 min), the tissue was stained with DAB, and nuclei were counterstained with hematoxylin.

### 2.7. Immunoblot assay

Harvested spinal cord tissues were homogenized on ice into radio-immunoprecipitation assay (RIPA)-based buffer (Wako, Osaka, Japan) [RIPA buffer containing 2 mM dithiothreitol (DTT), 1 mM sodium orthovanadate ($Na_3VO_4$), and complete protease inhibitor (Roche Diagnostics, Basel, Switzerland)]. After the tissue was centrifuged at 10,000 ×g, the supernatants were extracted. The total protein concentration was measured via the modified Bradford assay using BSA as a standard. Aliquots with 100 μg of protein were fractionated by 10% SDS polyacrylamide gel electrophoresis (SDS/PAGE), and the separated proteins were electrotransferred to polyvinylidene difluoride membranes using a transfer buffer. The membrane was probed with the following primary antibodies overnight at 4˚C: β-actin (A5316; Sigma-Aldrich) and NeuN (MAB377, Merch Millipore, Darmstadt, Germany). Subsequently, the membrane was incubated with HRP-conjugated anti-mouse immunoglobulin G (IgG) (GE Healthcare, Piscataway, NJ) or anti-rabbit IgG antibodies (GE Healthcare) for 1 h at room temperature. All antibodies were used according to the manufacturer's instructions. The membranes were stripped and reblotted twice to detect loading controls, as follows: 5 ml of stripping buffer (stripping buffer component: 10% SDS 40 ml, 1 M Tris HCl at pH 6.8 12.5 ml, and distilled water 146 ml) with 40 μl of β-mercaptoethanol added to the PVDF membrane. Then, the membrane was incubated at 50˚C for 30 min. All chemiluminescent signals were developed with enhanced chemiluminescence reagents (GE Healthcare).

### 2.8. Statistical analysis

All data are presented as means ± standard deviation. Statistical analysis was conducted using GraphPad Prism version 7.01, and a $P$ value of $<0.05$ was considered significant. Differences between the two groups were analyzed with unpaired Mann–Whitney U-tests. The Kruskal–Wallis H-test, followed by the Mann–Whitney U-test with Bonferroni correction, was used to compare between three or more groups.

## 3. Results

### 3.1. Pro-inflammatory cytokine expression in the spinal cord is significantly upregulated in a murine sepsis model

LPS was administered intraperitoneally to 10-week-old BALB/c male mice, and the serum IL-6 concentration was verified to be significantly elevated (Fig 1A). To investigate the inflammatory change induced in the spinal cord during sepsis, pro-inflammatory cytokine expression was analyzed by qRT-PCR. Fig 1B–1D shows that IL-1β, IL-6, and TNFα were drastically enhanced 2 h after LPS administration. Those upregulations normalized with time and returned to near baseline levels after 24 h, except for TNFα (Fig 1B–1D). Pro-inflammatory cytokine dynamics in the spinal cord were similar to the brain (Fig 1E–1G). In order to exclude the possibility of the effect of mice species, we performed the experiments in C57BL6 mice to find no difference between the two species (S1 Fig). Among inflammation-related genes other than pro-inflammatory cytokines, COX-2 was upregulated, but IL-36γ was not induced in the spinal cord (S2 Fig). The same study was performed with CLP mice as another representative

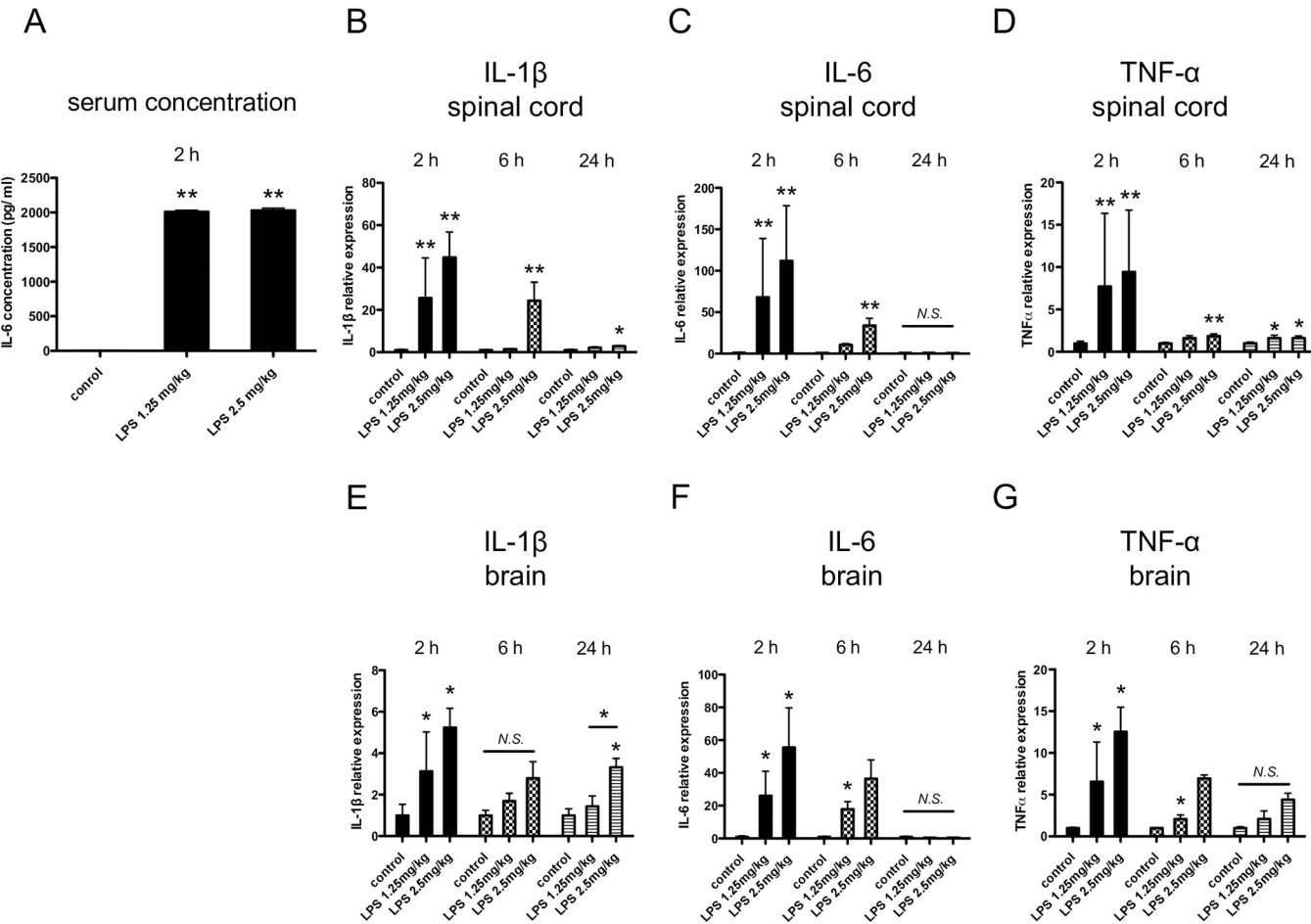

**Fig 1. Effect of intraperitoneal lipopolysaccharide (LPS) administration on pro-inflammatory cytokine expression in mouse spinal cords.** LPS 1.25 or 2.5 mg/kg, or the same amount of normal saline, was intraperitoneally administrated to 10-week-old BALB/c male mice. (A) Two-hour after the LPS administration, serum IL-6 concentration was determined with enzyme-linked immunosorbent assays (ELISA) (n = 4–5). mRNA expression levels of pro-inflammatory cytokines were determined in their spinal cords (B–D) and brains (E–G). 2, 6, and 24 h after LPS administration. Interleukin (IL)-1β (B and E), IL-6 (C and F), and tumor necrosis factor alpha (TNFα) (D and G) mRNA were assayed using real-time quantitative polymerase chain reactions (qRT-PCR; n = 4–7), and the expression levels were normalized to those of 18S rRNA and expressed relative to the mean in control mice. Data are presented as means ± standard deviations (S.D.); *P < 0.05 versus control; **P < 0.01 versus control; N.S., not significant.

model of sepsis. Spinal pro-inflammatory cytokine expression was significantly upregulated, and those inflammatory changes remained 24 h after the CLP operation (Fig 2).

## 3.2. Spinal morphological changes in LPS-treated mice

Histopathological analysis of the spinal cord sections stained with hematoxylin and eosin (HE) revealed that the morphology was almost normal in the control group (Fig 3). By contrast, the Th11 level of LPS-administered mice edematous change was confirmed especially in the anterior columns of LPS-treated mice so severe to be necrotic (Figs 3 and 4). In addition, densely stained ischemic neurons were widespread throughout the gray matter, especially around bilateral anterior horns (Fig 3). The nuclei of hyperchromatic cells were darker than the perikaryal, and the cytoplasm was shrunken (Fig 4). Despite these morphological changes in the spinal cord, infiltration of neutrophils and lymphocytes was not clearly observable under HE staining (Figs 3 and 4).

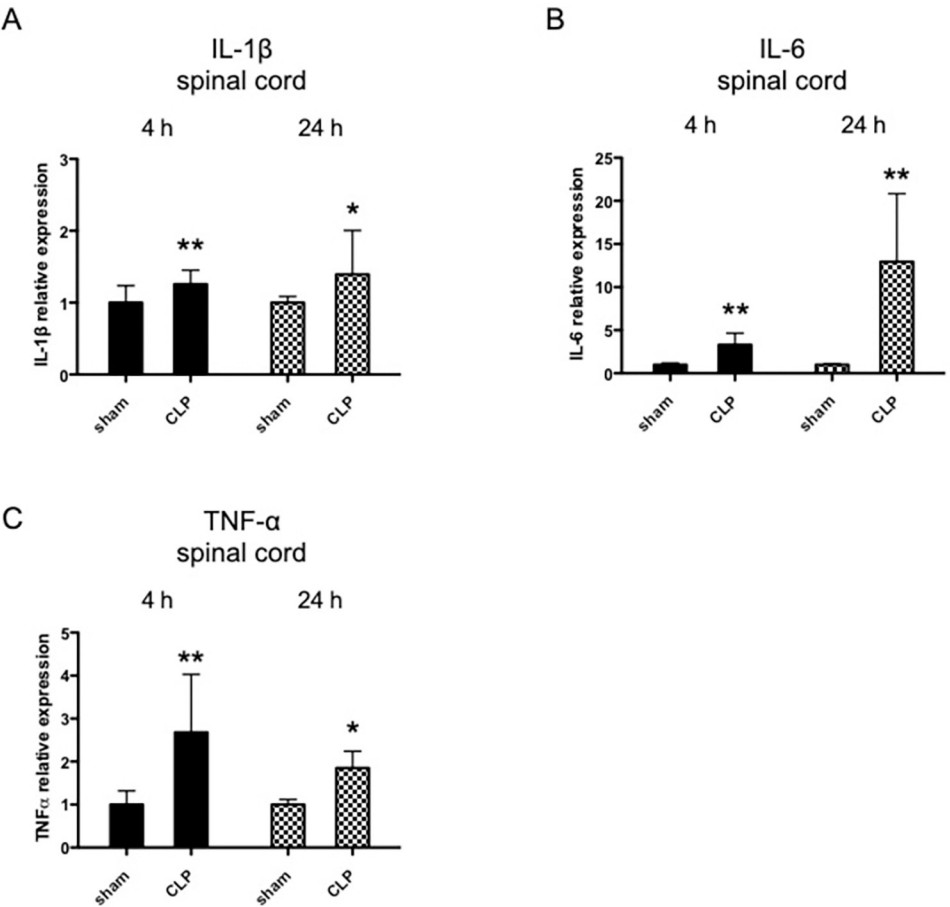

**Fig 2. Effect of cecal ligation puncture (CLP) on pro-inflammatory cytokine expression in mouse spinal cords.** CLP or sham operation was performed on 10-week-old BALB/c male mice under isoflurane anesthesia. mRNA expression levels of pro-inflammatory cytokines were determined in their spinal cords 4 and 24 h after the procedure. IL-6 (A), IL-1β (B), and TNFα (C) mRNA were assayed using real-time qRT-PCR (n = 4–6), and the expression levels were normalized to those of 18S rRNA and expressed relative to the mean in control mice. Data are presented as means ± S.D. *P < 0.05 versus control; **P < 0.01 versus control.

### 3.3. Spinal microglial activation in LPS-treated mice

Microglia, the resident immune cells in the CNS, are well known to release inflammatory cytokines. Thus, spinal microglia were analyzed using immunohistochemical staining for the microglia-specific protein IBA-1. The number of IBA-1-positive cells significantly increased 1 day after LPS administration and returned to its normal state 3 days after LPS administration (Fig 5A–5C). In contrast, the astrocyte-specific protein GFAP-positive cells were not quantitatively nor qualitatively changed after LPS administration (Fig 5A and 5D). NeuN-positive cells drastically decreased 3 days after LPS administration (Fig 5A and 5E), but NeuN expression was unchanged as revealed by immunoblotting (Fig 5F).

### 4. Discussion

This present study assessed the biochemical and histopathological changes to the spinal cord during sepsis. Systemic LPS administration is a minimally invasive, controllable, and reproducible method that mimics the acute phase of Gram-negative sepsis [21, 22]. This study

Control LPS 2.5 mg/kg

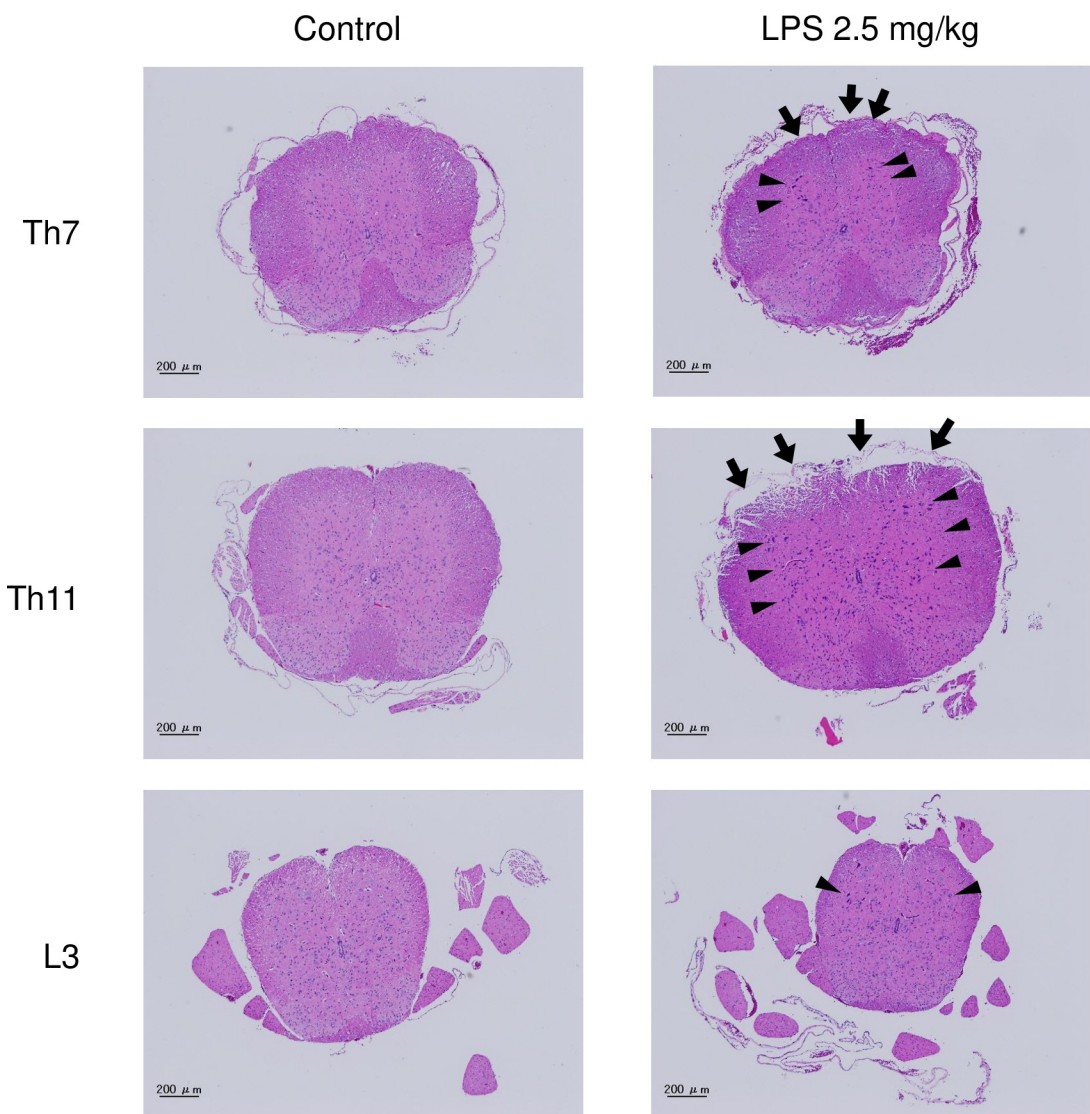

**Fig 3. Histological changes in spinal cords after intraperitoneal LPS administration.** LPS 2.5 mg/kg, or the same amount of normal saline, was intraperitoneally administrated to 10-week-old BALB/c male mice. The spinal cords were harvested 1 and 3 days after the administration, and Th 7, Th 11, and L3 transverse spinal sections were prepared and stained with hematoxylin and eosin (n = 4). A representative image of saline administrated mice was shown as control. Arrows indicate edematous changes of the anterior column and arrowheads depict densely stained cells. Scale bars, 200 μm (×100).

found that pro-inflammatory cytokines, including IL-1β, IL-6, and TNF-α, were drastically induced in the spinal cord as early as 2 h after LPS administration and almost alleviated within 24 h. This timeline is similar to that of the brain. Histopathological examination revealed that IBA-1-positive cells significantly proliferated. By contrast, no significant changes were observed in astrocytes. HE staining showed no infiltration of neutrophils and lymphocytes into the spinal cord. Actually, the expression of IL-36γ, induced by LPS in neutrophils [23], did not change in the spinal cord. Microglia are the major source of pro-inflammatory cytokines in the CNS in various neuroinflammatory diseases [24, 25]. In the current study, the largest induction of inflammatory cytokines occurred only 2 h after LPS administration; therefore, this reaction occurred within hours and was mainly caused by CNS residual microglia. On the

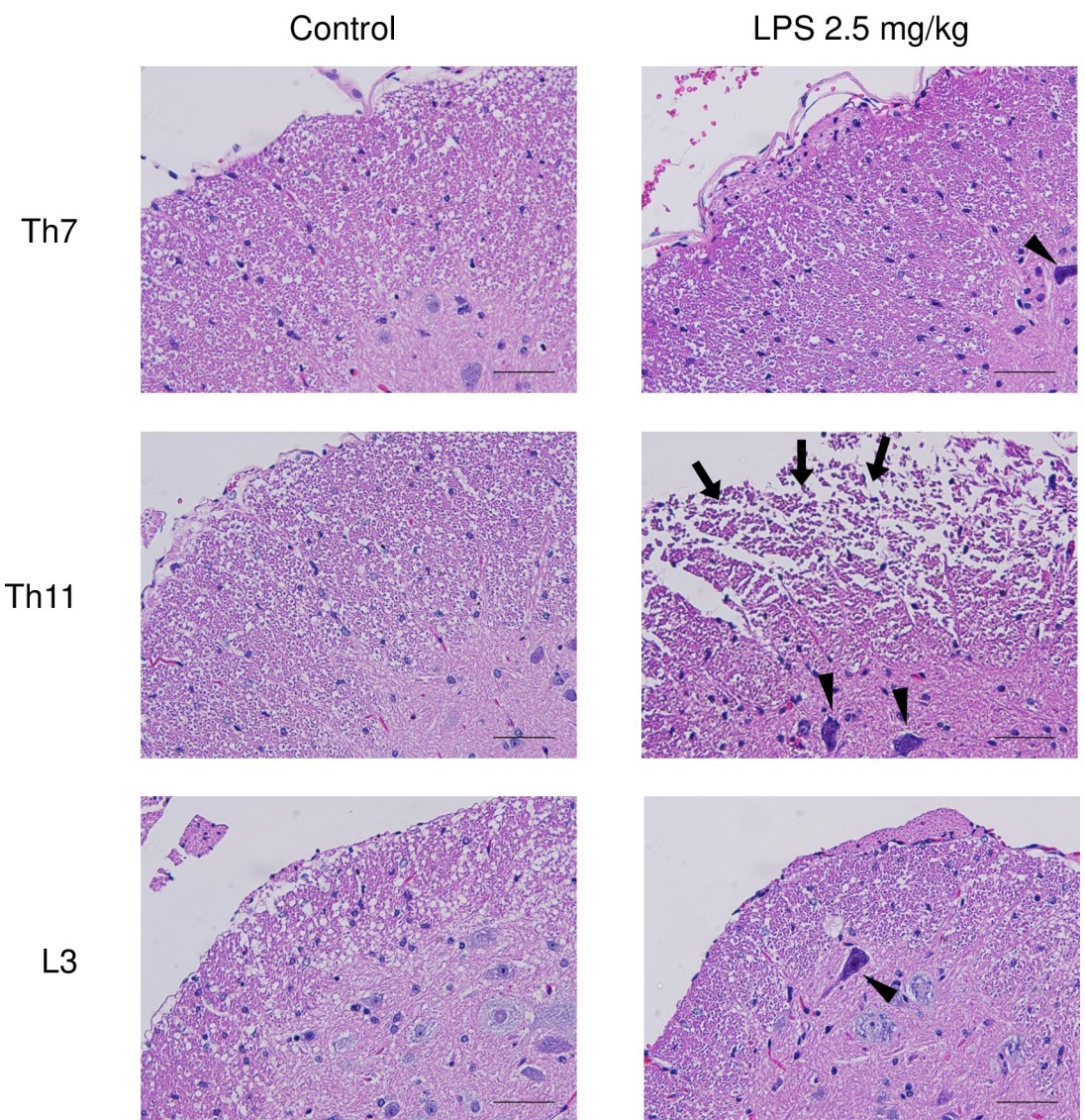

**Fig 4. Histological changes in spinal cords after intraperitoneal LPS administration.** High-magnification images of Fig 3 are shown. Arrows indicate edematous changes of the anterior column and arrowheads depict densely stained cells. Scale bars, 50 μm (×400).

other hand, peripheral inflammatory cells including monocyte-derived macrophage and neutrophils are reported to infiltrate into the CNS 24 h after LPS treatment [26, 27]. It is possible that peripheral cells are involved in the inflammatory changes that occurred afterward in the spinal cord. The degree of involvement of microglia and peripheral cells will need to be examined with more precise experiments like flow cytometry. Neuroinflammation is defined as the activation of the CNS's innate immune system in response to an inflammatory challenge and is characterized by a host of cellular and molecular changes within the CNS [28, 29]. Consequently, these findings strongly suggest that neuroinflammation occurs in the spinal cord and the brain during sepsis.

In the brain, previous reports have revealed that microglia are activated after LPS administration [30]. Inflammatory mediators, including pro-inflammatory cytokines, prostaglandins,

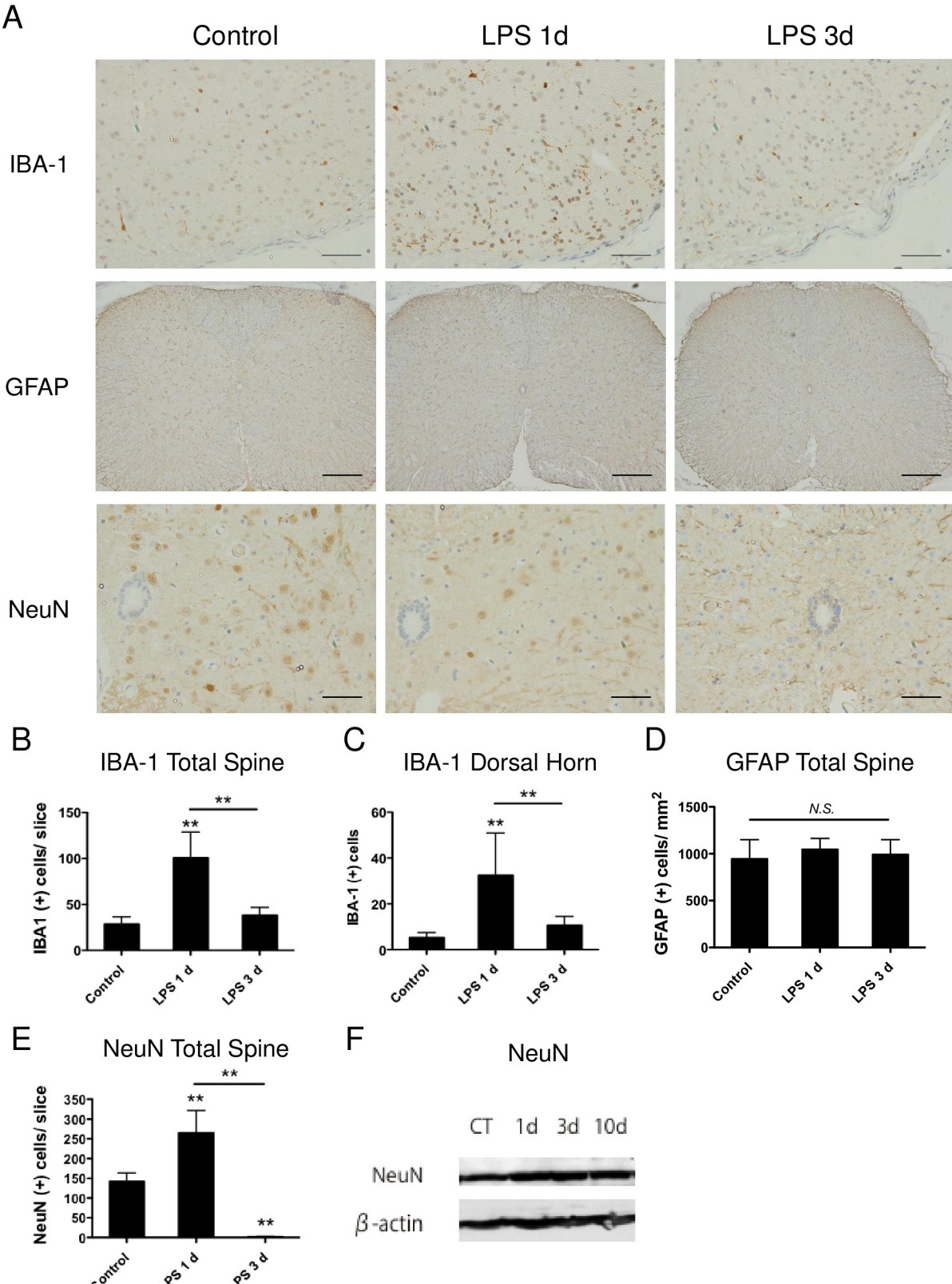

**Fig 5. Immunohistopathological changes in spinal cords after intraperitoneal LPS administration.** LPS 2.5 mg/kg, or the same amount of normal saline, was intraperitoneally administrated to 10-week-old BALB/c male mice and spinal cords were harvested 1 and 3 days after the

administration. Lower thoracic transverse spinal sections were immunostained for ionized calcium-binding adaptor molecule (IBA-1), glial fibrillary acidic protein (GFAP), and neuronal nuclei (NeuN) (A) and quantified (B–E) (n = 4). A representative image of saline administrated mice was shown as control. Scale bars, 200 μm (×100; IBA-1 and NeuN) and 50 μm (×400; GFAP). Data are presented as means ± S.D.; **P < 0.01, N.S., not significant. Thoracic and lumbar spinal cords of 10-week-old BALB/c mice 1 (1d), 3 (3d), and 10 (10d) days after LPS administration were analyzed with NeuN expression by immunoblot assay (F). CT; control. The figure is representative of three independent experiments.

and NOs, can impair the BBB and transmit signals to activate cerebral microglia [10–12]. In this study, the blood concentration of IL-6 was significantly increased. Therefore, it is possible that these peripherally derived inflammatory substances disrupted the BSCB, as the BSCB is similar to the BBB in function and morphology [19]. Importantly, the BSCB is reported to be more permeable than the BBB with relatively low expression of tight junction proteins [31]. Thus, cytokines can pass through the BSCB with more ease than the BBB [32]. However, in this study, the induction of pro-inflammatory cytokines in the spinal cord was synergistic with the brain. The conduction of the stimulus from the periphery to the CNS may be via nerves, cerebrospinal fluid, or blood flow, but the precise mechanisms need to be investigated.

Histopathological analysis of LPS-treated mouse spinal cords revealed serious neuronal ischemic changes as indicated by hyperchromatic neurons and severe edema in the white matter. These lesions were widespread from the thorax to the lumbar spinal cord. A previous study that examined the histopathological cerebral changes with a septic murine model showed similar neuronal hyperchromatic alterations [5]. The altered cerebral circulation due to systemic hypotension, thrombus formation, and impaired cerebral vasoreactivity may induce neuronal ischemia and degeneration [33]. Therefore, the histological changes in the spinal cord can be induced by the same mechanism as in the brain, although the precise mechanism needs to be elucidated. It is possible that the alteration of the spinal cord was extremely widespread and severe since white matter edema was partially necrotic. With such drastic pathological changes, the function of the spinal cord could be impaired. Conversely, neuromuscular weakness and acute diffuse muscle weakness in critically ill patients, called ICU-acquired weakness (ICU-AW), occur during sepsis [34–36]. ICU-AW occurs in 46% of severely septic patients [35, 37], and the most common causes are likely critical illness polyneuropathy (CIP), critical illness myopathy (CIM), and the overlap of critical illness polyneuromyopathy (CIPNM) [38]. However, the involvement of the spinal cord in neuromuscular weakness has not been investigated, although spinal motor neuron excitability was reduced in a CLP model rat [39]. Considering there are few effective treatments for CIP, CIM, and CIPNM [40], the clinical impact could be significant if the involvement of spinal cord lesions is confirmed and therapeutic agents that target spinal microglia such as minocycline are effective. Therefore, it is necessary to clarify the extent of spinal cord involvement in sepsis-induced neuromuscular weakness.

One of the limitations of this study is that the animal model used is an LPS administration model. Sepsis is a very complex condition, and it is quite possible that the LPS model and actual sepsis may not always match. In fact, in another model utilized, i.e., CLP, the induction of inflammatory cytokines in the spinal cord was milder and lasted longer than the LPS model. Additionally, only short-term investigation of LPS administration, i.e., within 72 h, was conducted. This was due to the quick loss the LPS effect. During actual sepsis, the time course is usually longer, so it is necessary to observe animals over the longer time course. In addition, NeuN-positive cells were drastically decreased in LPS-treated mice. However, according to a previous report, loss of NeuN immunoreactivity after cerebral ischemia does not indicate neuronal cell loss [41]. In this study, the expression level of NeuN was not changed with LPS in immunoblotting. Therefore, as quantification of NeuN-positive cells might be influenced by

inflammation-induced ischemia, other neuronal markers including MAP2, synaptophysin, and PSD95 may be more preferable. Moreover, the edematous change was confirmed by HE staining of the anterior columns. Therefore, immunohistochemical examination for neuronal network using chondroitin sulfate proteoglycans (CSPGs) may be useful. Finally, in sepsis, hypotension and accompanying increase in lactate level can occur [42, 43]. Therefore, in this study, secondary hemodynamic changes after LPS administration may have affected ischemic neuronal changes in the spinal cord. It may be necessary to investigate the cause of ischemic changes in the spinal cord in more detail by analyzing secondary changes including blood pressure and oxygenation.

In summary, inflammatory changes in the spinal cord were examined using a murine sepsis model. After LPS administration, drastic induction of inflammatory cytokines and a marked increase of activated microglia were observed in the spinal cord. Histologically, extensive edema and necrosis of the white matter and ischemic changes in the gray matter were observed. These findings strongly suggest that inflammatory changes occurred in the spinal cord during sepsis and that those changes may contribute to ICU-AW.

## Supporting information

**S1 Fig. Effect of intraperitoneal lipopolysaccharide (LPS) administration on pro-inflammatory cytokine expression in C57BL6 mouse spinal cords.** LPS 1.25 or 2.5 mg/kg, or the same amount of normal saline, was intraperitoneally administrated to 10-week-old C57BL6 male mice. mRNA expression levels of pro-inflammatory cytokines were determined in their spinal cords (A-C). mRNA were assayed using real-time quantitative polymerase chain reactions (qRT-PCR; n = 3–5), and the expression levels were normalized to those of 18S rRNA and expressed relative to the mean in control mice. Data are presented as means ± standard deviations (S.D.); $^*$P < 0.05 versus control; N.S., not significant.
(TIF)

**S2 Fig. Effect of intraperitoneal lipopolysaccharide (LPS) administration on pro-inflammatory cytokine cyclooxygenase (COX)-2 and interleukin (IL)-36γ expression in mice spinal cords.** LPS 2.5 mg/kg, or the same amount of normal saline was intraperitoneally administrated to 10-week-old C57BL6 male mice. Pro-inflammatory cytokine COX-2 (A) and IL-36γ (B) mRNA expression were determined in their spinal cords 2 h after LPS administration. MRNA were assayed using real-time quantitative polymerase chain reactions (qRT-PCR; n = 4), and the expression levels were normalized to those of 18S rRNA and expressed relative to the mean in control mice. Data are presented as means ± standard deviations (S.D.); $^*$P < 0.05 versus control; N.S., not significant.
(TIF)

**S1 File.**
(XLSX)

## Acknowledgments

We wish to thank Center for Anatomical, Pathological and Forensic Medical Researches, Graduate School of Medicine, Kyoto University for technical help for immunocytochemical analysis and advice.

## Author Contributions

**Conceptualization:** Tomoharu Tanaka.

**Investigation:** Akiko Hirotsu, Mariko Miyao, Tomoharu Tanaka.

**Methodology:** Akiko Hirotsu, Kenichiro Tatsumi, Tomoharu Tanaka.

**Project administration:** Akiko Hirotsu.

**Resources:** Mariko Miyao.

**Supervision:** Tomoharu Tanaka.

**Writing – original draft:** Akiko Hirotsu.

**Writing – review & editing:** Tomoharu Tanaka.

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
