## [Decision Letter · Decision Letter 0]

11 Feb 2022

PONE-D-21-28822Sepsis-Associated Neuroinflammation in the Spinal CordPLOS ONE

Dear Dr. 

Thank you for submitting your manuscript to PLOS ONE. After careful consideration, we feel that it has merit but does not fully meet PLOS ONE’s publication criteria as it currently stands. Therefore, we invite you to submit a revised version of the manuscript that addresses the points raised during the review process.

We look forward to receiving your revised manuscript.

Kind regards,

Rosanna Di Paola, MD

Academic Editor

PLOS ONE

Journal Requirements:

In your cover letter, please note whether your blot/gel image data are in Supporting Information or posted at a public data repository, provide the repository URL if relevant, and provide specific details as to which raw blot/gel images, if any, are not available. Email us at plosone@plos.org if you have any questions

3. Please provide additional information within the Methods section regarding steps taken to alleviate suffering of the animals during the study. Furthermore , please provide a justification for the sample size used in your study, including any relevant power calculations (if applicable). And finally please indicate the timepoint at which the animals were scarified

4. "Thank you for stating the following in your Competing Interests section:  

"No"

 This information should be included in your cover letter; we will change the online submission form on your behalf."

Reviewers' comments:

Reviewer's Responses to Questions

**Comments to the Author**

1. Is the manuscript technically sound, and do the data support the conclusions?

Reviewer #1: Yes

2. Has the statistical analysis been performed appropriately and rigorously? 

Reviewer #1: No

3. Have the authors made all data underlying the findings in their manuscript fully available?

Reviewer #1: Yes

4. Is the manuscript presented in an intelligible fashion and written in standard English?

Reviewer #1: Yes

5. Review Comments to the Author

Reviewer #1: The manuscript investigated the impact of sepsis on the spinal cord and further presented the neuroinflammation of this tissue with a focus on pro-inflammatory cytokines and microglia. Based on an original idea, this article introduces potential new area of investigations into the effects of sepsis on central nervous system.

Major concerns:

- An isolation of microglia followed by qRT-PCR would provide a real added value using specific markers of inflammation or ROS. Moreover, the absence of neutrophils is surprising especially if the BSCB is broken down by LPS as expected. These leukocytes have already been observed in spinal cord using infectious models. Defined by a cytokine storm, the administration of LPS should induce an infiltration of these cells within 24 hours.

- Difficult to compare data obtained from an endotoxemic model using a relative low dose of LPS (with a high survival rate at 5 days) and a lethal CLP model (100 % of death within 72 hours). To compare your sepsis models, a puncture using a 23 or 25-gauge needle would be much better.

- Neuroinflammation in the spinal cord is recognized following infections and was expected after endotoxin injection. An analyze of more specific markers of the spinal cord such as CSPG would be positive.

- The effects of LPS on arterial pressure and lactate concentration should also be included in the article to confirm the sepsis model. One or two references would also get the job done.

Additional concerns:

- An analysis of the macrophages derived from monocytes would be interesting.

- 78: Lipopolyssacharide from E.Coli 055:B5 is designated as L28880 instead of L2880.

- 146: The target of anti-NeuN and anti-IBA1 antibodies should be explained directly after the first mention.

- 185-194: I have some concerns about statistics:

• Was an analysis of variance done on data families to avoid unsuitable comparisons?

• “Sample size was determined on the basis of a power analysis” -> What is the result of this power analysis?

• “Normality was tested using data from our recent similar studies” -> How can you use data not involved in this study to analyze normality of your results?

- 220: “there were no peripheral inflammatory cell infiltrations …. “: This sentence is too affirmative.

- 252: "it is natural to think that the inflammatory cytokines are mainly derived from microglia, although vascular endothelial cells may be another source of production”. Other cells are also implicated in inflammatory cytokines production. This sentence is also too affirmative.

- 268: “in our study, the induction of pro-inflammatory cytokines in the spinal cord was very synergistic with the brain, suggesting that there may be a mechanism for transmitting inflammatory stimuli from the spinal cord to the brain or vice versa”. How can you observe a mechanism transmitting inflammatory stimuli from the brain to the spinal cord if the BSCB is reported to be more permeable than the BBB and without BBB dysfunction observed?

6. PLOS authors have the option to publish the peer review history of their article (what does this mean?). If published, this will include your full peer review and any attached files.

Reviewer #1: No

---

## [Author Response · Author response to Decision Letter 0]

15 Mar 2022

Dear Dr. Rosanna Di Paola

For possible publication in PLOS ONE, we wish to re-submit revised version of the manuscript entitled “Sepsis-Associated Neuroinflammation in the Spinal Cord” (PONE-D-21-28822) We substantially revised the manuscript in response to the comment of the editor and reviewer to address some of the issues raised in the evaluation of the original manuscript. 

We wish to express our strong appreciation to the editor and reviewer for his or her insightful comments on our paper. We feel the comments have helped us significantly improve the paper.

The reviewer’s comments are indicated in point 8 bold face letter followed by our responses in point 10 letter.

Comments from the reviewer 

Major points:

1. An isolation of microglia followed by qRT-PCR would provide a real added value using specific markers of inflammation or ROS. Moreover, the absence of neutrophils is surprising especially if the BSCB is broken down by LPS as expected. These leukocytes have already been observed in spinal cord using infectious models. Defined by a cytokine storm, the administration of LPS should induce an infiltration of these cells within 24 hours.

In this study, we mainly focus on whether sepsis induces neuroinflammation in the spinal cord or not. As the reviewer pointed out, we have to admit that we did not clarify the mechanism of inflammation in the spinal cord well, for example, what category of inflammatory cells are involved. In order to clarify that point, we added the RT-PCR experiment with another representative inflammation-related gene, COX-2 and neutrophil-specific gene, IL-36g to find that COX-2 was upregulated, but IL-36g was stable with LPS administration in the spinal cord (Supplementary figure). According to the past report, infiltration of peripheral inflammatory cells occurs within 24 hours of LPS administration, but, in our study, upregulation of inflammatory cytokines was found as early as 2 hours after LPS treatment. So, we think, at least, for the neuroinflammation observed at least within a few hours after LPS administration, the contribution of peripheral inflammatory cells may be small. The results of supplementary figure　are not definitive, but support that point. However, the involvement of these cells cannot be ruled out over a longer period of time. We have to admit that our data is not enough to conclude that what cells are main contributor of neuroinflammation in the spinal cord, and more detailed investigation will in need. We substantially revised the manuscript, especially in the 1st paragraph of discussion section in order to escape the misleading.

2) Difficult to compare data obtained from an endotoxemic model using a relative low dose of LPS (with a high survival rate at 5 days) and a lethal CLP model (100 % of death within 72 hours). To compare your sepsis models, a puncture using a 23 or 25-gauge needle would be much better.

In the preliminary experiment, we examined the CLP model with 24G needle, but systemic inflammation including elevation of plasma IL-6 did not occur. Therefore, we adopted 18G in our experiments. However, as the reviewer pointed out, the degree of inflammation in the spinal cord was very different between the two models. In the current study, we focused on the occurrence of neuroinflammation in the spinal cord, and conclude that happens with the finding of proinflammatory cytokine induction in both septic models. But, the CLP model should be examined more precisely in the future.

3) Neuroinflammation in the spinal cord is recognized following infections and was expected after endotoxin injection. An analyze of more specific markers of the spinal cord such as CSPG would be positive.

As the reviewer pointed out, immunohistochemical examination using specific markers of the spinal cord would be more informative, especially focusing on spinal neuronal networking. In the discussion section of revised manuscript (line327-330), we added the sentences as follows;

“Moreover, the edematous change was confirmed by HE staining of the anterior columns. Therefore, immunohistochemical examination for neuronal network using chondroitin sulfate proteoglycans (CSPGs) may be useful.”

Other comments:

4) The effects of LPS on arterial pressure and lactate concentration should also be included in the article to confirm the sepsis model. One or two references would also get the job done

As the reviewer pointed out, the secondary changes including hypotension and lactate accumulation induced by systemic inflammation in sepsis can affect spinal cord, especially its ischemic changes. Thus, we added the sentences in the discussion section (line330-335) as follows;

“Finally, in sepsis, hypotension and accompanying increase in lactate level can occur [42, 43]. Therefore, in this study, secondary hemodynamic changes after LPS administration may have affected ischemic neuronal changes in the spinal cord. It may be necessary to investigate the cause of ischemic changes in the spinal cord in more detail by analyzing secondary changes including blood pressure and oxygenation.”

Additional concerns:

- An analysis of the macrophages derived from monocytes would be interesting.

The reviewer’s suggestion is important. In the current study, we could not clarify the point, but we agree that point should be precisely examined in the future. In the 1st paragraph of discussion section, we described about that point. 

- 78: Lipopolyssacharide from E.Coli 055:B5 is designated as L28880 instead of L2880.

- 146: The target of anti-NeuN and anti-IBA1 antibodies should be explained directly after the first mention 

We corrected the description as the reviewer indicated.

- 185-194: I have some concerns about statistics:

• Was an analysis of variance done on data families to avoid unsuitable comparisons?

• “Sample size was determined on the basis of a power analysis” -> What is the result of this power analysis?

• “Normality was tested using data from our recent similar studies” -> How can you use data not involved in this study to analyze normality of your results?

The reviewer has raised an important indicate. In the preliminary experiment of proinflammatory cytokine upregulations in the spinal cord of LPS treated mice, we performed normality test to find the data was normally distributed. In addition, power analysis using 1.2 fold of the upregulation of cytokines as significant was performed. However, considering other experiments, it is practically difficult to perform normality test and power analysis. So, in order to avoid misunderstanding, we reexamined all data with non-parametric analysis. Thus, we described in the statistics section of the revised manuscript as follows;

“Differences between the two groups were analyzed with unpaired Mann–Whitney U-tests. The Kruskal–Wallis H-test, followed by the Mann–Whitney U-test with Bonferroni correction, was used to compare between three or more groups.”(line 195-201)

- 220: “there were no peripheral inflammatory cell infiltrations …. “: This sentence is too affirmative.

We agree the reviewer’s opinion about that point, so we redescribed that part as follows

”Despite these morphological changes in the spinal cord, infiltration of neutrophils and lymphocytes was not clearly observable under HE staining (Figs. 3 and 4).” (line230-232)

- 252: "it is natural to think that the inflammatory cytokines are mainly derived from microglia, although vascular endothelial cells may be another source of production”. Other cells are also implicated in inflammatory cytokines production. This sentence is also too affirmative.

We agree the reviewer’s opinion about that point, so we redescribed that part as follows 

“It is possible that peripheral cells are involved in the inflammatory changes that occurred afterward in the spinal cord. The degree of involvement of microglia and peripheral cells will need to be examined with more precise experiments like flow cytometry.” (revised manuscript line264-267)

- 268: “in our study, the induction of pro-inflammatory cytokines in the spinal cord was very synergistic with the brain, suggesting that there may be a mechanism for transmitting inflammatory stimuli from the spinal cord to the brain or vice versa”. How can you observe a mechanism transmitting inflammatory stimuli from the brain to the spinal cord if the BSCB is reported to be more permeable than the BBB and without BBB dysfunction observed?

In line with the reviewer’s comment, we deleted that part in the revised manuscript.

Finally, in addition to the reviewer’s comments, we revised the manuscript as follows;

1) In the original manuscript, we described all experiments were performed with C57BL6, but it was clerical error. We performed most experiments with BALB/c mice, but used C57BL6 mice for supplementary experiments, as BALB/c 10 week male mice were transiently difficult to purchase. As show in supplementary figure 1, the main finding of this study, LPS-induced proinflammatory cytokine upregulation, was common in C57BL6 as well as BALB/c mice.

2) We correctly rewritten the number of specimens.

We really appreciate the reviewer for giving us the opportunity to strengthen our manuscript with your valuable comments and queries. We trust that the revised manuscript is suitable for publication. 

Sincerely yours,

Tomoharu Tanaka, MD. PhD.

Department of Anesthesia, Kyoto University Hospital, Sakyo-ku, Kyoto 606-8507, Japan.

Tel:+81-75-751-3436,Fax:+81-75-752-3259

Mail to:665tana@kuhp.kyoto-u.ac.jp

---

## [Decision Letter · Decision Letter 1]

1 Jun 2022

Sepsis-Associated Neuroinflammation in the Spinal Cord

PONE-D-21-28822R1

Dear Dr. 

We’re pleased to inform you that your manuscript has been judged scientifically suitable for publication and will be formally accepted for publication once it meets all outstanding technical requirements.

Kind regards,

Rosanna Di Paola, MD

Academic Editor

PLOS ONE

Additional Editor Comments (optional):

Reviewers' comments:

Reviewer's Responses to Questions

**Comments to the Author**

1. If the authors have adequately addressed your comments raised in a previous round of review and you feel that this manuscript is now acceptable for publication, you may indicate that here to bypass the “Comments to the Author” section, enter your conflict of interest statement in the “Confidential to Editor” section, and submit your "Accept" recommendation.

Reviewer #2: All comments have been addressed

2. Is the manuscript technically sound, and do the data support the conclusions?

Reviewer #2: Yes

3. Has the statistical analysis been performed appropriately and rigorously? 

Reviewer #2: Yes

4. Have the authors made all data underlying the findings in their manuscript fully available?

Reviewer #2: Yes

5. Is the manuscript presented in an intelligible fashion and written in standard English?

Reviewer #2: Yes

6. Review Comments to the Author

Reviewer #2: the authors have fully addressed all the concerns raised by the reviewer anf therefoe the manuscript is now suitable for publciation

7. PLOS authors have the option to publish the peer review history of their article (what does this mean?). If published, this will include your full peer review and any attached files.

Reviewer #2: No

---

## [Editor Report · Acceptance letter]

3 Jun 2022

PONE-D-21-28822R1 

Sepsis-Associated Neuroinflammation in the Spinal Cord 

Dear Dr. Tanaka:

I'm pleased to inform you that your manuscript has been deemed suitable for publication in PLOS ONE. Congratulations! Your manuscript is now with our production department. 

Kind regards, 

on behalf of

Dr. Rosanna Di Paola 

Academic Editor

PLOS ONE